# Identification of tolerance levels on the cold-water coral *Desmophyllum pertusum (Lophelia pertusa)* from realistic exposure conditions to suspended bentonite, barite and drill cutting particles

**Thierry Baussant**[1]*, **Maj Arnberg**[1¤a], **Emily Lyng**[1], **Sreerekha Ramanand**[1], **Shaw Bamber**[1], **Mark Berry**[1¤b], **Ingrid Myrnes Hansen**[2,3], **Dick Van Oevelen**[4], **Peter Van Breugel**[4]

**1** NORCE Norwegian Research Centre AS, Randaberg, Norway, **2** Ecotone AS, Trondheim, Norway, **3** Norwegian University of Science and Technology, Trondheim, Norway, **4** Department of Estuarine and Delta Systems, NIOZ – Royal Netherlands Institute for Sea Research, Yerseke, the Netherlands

¤a Current address: Akvaplan-niva, Pirsenteret, Trondheim, Norway
¤b Current address: Marine Scotland, Fishery Office, Shetland, United Kingdom
* thba@norceresearch.no

**Data Availability Statement:** The data used in the paper are now uploaded in Dryad datadryad.org/

## Abstract

Cold-water coral (CWC) reefs are numerous and widespread along the Norwegian continental shelf where oil and gas industry operate. Uncertainties exist regarding their impacts from operational discharges to drilling. Effect thresholds obtained from near-realistic exposure of suspended particle concentrations for use in coral risk modeling are particularly needed. Here, nubbins of *Desmophyllum pertusum* (*Lophelia pertusa*) were exposed shortly (5 days, 4h repeated pulses) to suspended particles (bentonite BE; barite BA, and drill cuttings DC) in the range of ~ 4 to ~ 60 mg.l$^{-1}$ (actual concentration). Physiological responses (respiration rate, growth rate, mucus-related particulate organic carbon OC and particulate organic nitrogen ON) and polyp mortality were then measured 2 and 6 weeks post-exposure to assess long-term effects. Respiration and growth rates were not significantly different in any of the treatments tested compared to control. OC production was not affected in any treatment, but a significant increase of OC:ON in mucus produced by BE-exposed (23 and 48 mg.l$^{-1}$) corals was revealed 2 weeks after exposure. Polyp mortality increased significantly at the two highest DC doses (19 and 49 mg.l$^{-1}$) 2 and 6 weeks post-exposure but no significant difference was observed in any of the other treatments compared to the control. These findings are adding new knowledge on coral resilience to short realistic exposure of suspended drill particles and indicate overall a risk for long-term effects at a threshold of ~20 mg.l$^{-1}$.

stash/dashboard with DOI https://doi.org/10.5061/dryad.t76hdr827.

**Funding:** This study and the writing of the manuscript was financed by Equinor AS under contract #4590073409. We are particularly grateful for their role in the study design, discussions of the results, decision to publish, discussions and comments on the manuscript made by Tone K. Frost and Ingunn Nilsen from Equinor AS. The funders had no additional role in study design, data collection and analysis, or preparation of the manuscript.

**Competing interests:** This study and the writing of the manuscript was financed by Equinor AS. There are no patents, products in development or marketed products associated with this research to declare. This does not alter our adherence to PLOS ONE policies on sharing data and materials.

# Introduction

Cold-water corals (CWC), and the habitats they represent for a diverse marine life, have survived and expanded for thousands of years in the deep-sea. However, in the recent Anthropocene period, their fate is now threatened by global and regional anthropogenic activities with several consequences for their survival [1–5]. Among CWC, the scleractinian *Lophelia pertusa* (Linnaeus, 1758), now formerly synonymized to *Desmophyllum pertusum* (Linnaeus, 1758) [6], is widely distributed across a wide depth range on the Norwegian continental shelf [7, 8]. This hard-bottom species is forming reefs, true hotspots of biodiversity in the deep ocean, but is listed as "near-threatened" [9] mainly because of extended physical damages by bottom trawling [1, 10, 11]. The exploration for petroleum activities adds another potential hazard to these hotspots of life [3, 12–14]. The wastes generated by these activities are dominated by DC and drilling fluids/muds. DC are particles of crushed rock from the formation being drilled. An exploration well usually takes about 1 to 3 months to drill [15]. However, actual drilling occurs only 30 to 50% of the time during a typical exploratory drilling operation. The chemical and mineral composition of cuttings reflects geochemistry of the rock formation [16–18]. During drilling of the near-surface, upper well sections, cuttings and associated mud are deposited directly on the sea floor at the drill site or are pumped through a cuttings transport system (CTS) away from the platform to a site more remote from sensitive seafloor habitats [15]. Drilling mud solids (2–10%) are associated with cuttings and discharged to the sea [18]. Water based drilling muds (WBM) are mixtures of fine-grained solids, inorganic salts, and organic compounds dissolved or dispersed in water. WBM are considered not harmful to the marine environment and are defined as PLONOR chemicals (Pose Little Or NO Risk to the Environment) according to OSPAR [19]. Therefore, WBM and associated cuttings are normally permitted for discharge to offshore waters in most countries, included Norway, through environmental discharge permits dependent of the local marine habitats in the area of interest. WBM consist of water and fine-grained solids such as bentonite (BE) clay added to the drilling mud to maintain viscosity, and barite (BA), used as weighting agent in drilling muds. On the Norwegian continental shelf (NCS), there are periodic bulk discharges of small amounts of WBM during drilling and a bulk discharge of a larger volume of WBM at the end of drilling and petroleum operators must obtain a permit from the Norwegian Environment Agency (NEA) under the regulation of the Pollution Control Act (section § 68 for Discharge of cuttings, sand and other solid particles). Guidelines [20] for environmental monitoring of petroleum activities on the NCS are provided to support and fulfil the regulatory requirements and a handbook was recently prepared [21] to recommend methods for baseline and visual mapping prior to exploration drilling in areas potentially housing vulnerable fauna such as CWC.

Coarse rock cuttings that deposit at the bottom and smaller suspended particles in the water column following mobilization by offshore activities can potentially impact the corals [14, 22]. Tolerance values for natural and anthropogenic sediment (such as from dredging) disturbances, including resuspended sediment particles, have been established for shallow water coral species, with biological functions affected such as sediment rejection, mucus production and polyp activity [23–25]. Less is known for CWC but recently, recommended levels based on studies for burial and smothering from drill particle sedimentation [26, 27] were proposed and are now used by offshore industries. Regarding suspended drill particles, the scientific information is scarcer and risk for the corals are mostly based on physical parameters such as turbidity and current measurements [21]. A few studies in the laboratory have shed new insight in effects to corals exposed to suspended drilling particles in the range of 2–52 mg. l$^{-1}$ (actual concentration), and concluded overall that several coral responses are observed at or above a level of ~ 10 mg.l$^{-1}$ [28, 29]. However, this level should be considered conservative as

these studies used continuous or repeated pulses of drill particle exposure over long period (2.5 to 12 weeks) that differ from those typical for actual field situation. Harris et al. [30] emphasize the critical importance of defining and testing realistic and environmentally relevant exposure scenarios and to comprehensively justify those exposure conditions. Browne et al. [23] found that periodic suspended sediment exposure was less detrimental to shallow-water corals than constant exposure. In the field, drill particle exposures are fluctuating and characterized by plume of particles over short periods (few days) alternating with periods of no discharges [15]. Exposure studies mimicking more realistic field exposure of drill particles are missing. Two recent studies on effects of suspendend drilling particles to different larval life stages of *D. pertusum* concluded that following short 24h exposure, the concentration at which 50% of 8 days and 21-days old larvae showed behavioural effects (EC50) was observed at ~ 10 mg.l$^{-1}$, 20 mg.l$^{-1}$and 40 mg.l$^{-1}$, respectively for BE, BA and DC [31], showing overall a greatest sensitivity to BE compared to BA and DC exposures. Post-exposure recovery rates were also the lowest for BE. The authors explained this difference by the "stickiness" of BE particles to the larva, which cannot recover as it is incapable of freeing itself of the mucus capsule.

The primary objective of the present research was to obtain effect concentration thresholds using realistic short exposure scenarios of suspended drilling material to nubbins of the coral *D. pertusum*. Pulses of drill particle exposure for DC, BA and BE were repeated over 5 days and then the long-term effects to corals measured 2 and 6 weeks post-exposure (recovery in natural seawater) using coral biological end-points that previously showed effects in long-term exposure in the laboratory. This included physiological end-points (respiration, growth and coral mucus production) and polyp mortality [28, 32]. The general hypothesis was that short-term exposure to suspended drill particles increases tolerance effect threshold measured previously in long-term DC exposure at ~10 mg.l$^{-1}$. Also, from the above-mentioned coral larvae studies, we hypothesized that BE particles elicit long-term effects at lower concentration than BA and DC. The endpoints measurements allowed to define coral threshold concentrations from field-realistic exposure to suspended drill particles to use in coral risk assessment by offshore managers.

## Materials and methods

### Collection and maintenance of corals

*D. pertusum* is not defined as a threatened or protected species in Norway, but due to their reported decline, they are defined as "nearly-threatened" according to the terms used in the Norwegian Red List for Ecosystems and Habitats [9]. Collection and maintenance of the corals in the laboratory were performed according to recommendations and operational practices applied by CWC scientists working experimentally with these organisms [33]. The sampling of the corals was conducted outside of national parks or other type of protected areas, so that no specific permissions were required. White and orange morphs of *D. pertusum* colonies were collected in April 2018 with a Remote Operated Vehicle (Sperre Subfighter 7500 "Minerva" ROV, NTNU, Norway) aboard R/V Gunnerus in the Trondheimsfjorden (Norway) near the Tautra reef (63.5749˚N 10.6055˚E) where their density is high, and they can be collected from relatively shallow depths. Temperature difference in sea water from depth of collection to surface did not exceed 2˚C as was air temperature difference. The corals were placed in buckets pre-filled with water collected at 80m depth in the fjord and kept in dark until return to shore. Corals were then transported in refrigerated water by air cargo from Trondheim to the NORCE marine facility (Mekjarvik, Norway), where the experimental work was carried out. The water temperature was controlled and stable during transport to the facility. In the NORCE facility, the corals were maintained in flow-through fjord water (~1000ml.min$^{-1}$) and

allowed to acclimate for three weeks in conditions reported in Baussant et al. [34] (kept in darkness with running seawater from Byfjord (Rogaland, Norway) collected at 80 m depth, 7.5 ± 0.2°C, 33 ± 0.5 PSU, and fed at least twice a week with a solution of live *Artemia salina* nauplii (stock concentration ~ 8000 ml$^{-1}$; ~10 ml per feeding event). During this period, coral welfare was evaluated mainly from the visual observation of polyp tentacles expansion. There was no visible excessive mucus secretion nor mortality. Coral individuals with most of the polyps frequently retracted in the skeleton during these observations were considered unhealthy and were not used further in the experimental study. Only white morphotype coral individuals were used.

## Experimental design

Two experiments were performed consecutively. First, an experiment with field-collected drill cutting material (referred as DC experiment) was conducted from June 2018 –August 2018. This was followed by an experiment with barite (BA) and bentonite clay (BE) particles (December 2018 –February 2019, herein referred as BABE experiment).

The overall setup and design was as described in Baussant et al. [28] with few modifications, and the sampling plan was the same in both experimental studies for the measurements of all end-points parameters. Coral nubbins were transferred into individual 15 L plexiglass cone-shaped containers (hereafter referred as "cones") supplied with 220 ±30 ml.min$^{-1}$ sand-filtered and temperature-regulated water from Byfjord. There, the corals were acclimatized another 4 weeks under these conditions before the exposure started. There were 3 replicate cones per treatment concentration and for the control, as shown in Fig 1. Further, there were 6 to 8 nubbins per cone. In each of them, three (BABE experiment; 6±2 polyps.nubbin$^{-1}$) to four (DC experiment; 5±1 polyps.nubbin$^{-1}$) nubbins were used for growth and respiration measurements and to determine polyp mortality. For POM-based mucus release, 3 (BABE experiment; 11±5 polyps.nubbin$^{-1}$) to 4 (DC experiment; 14±5 polyps.nubbin$^{-1}$) other nubbins were added

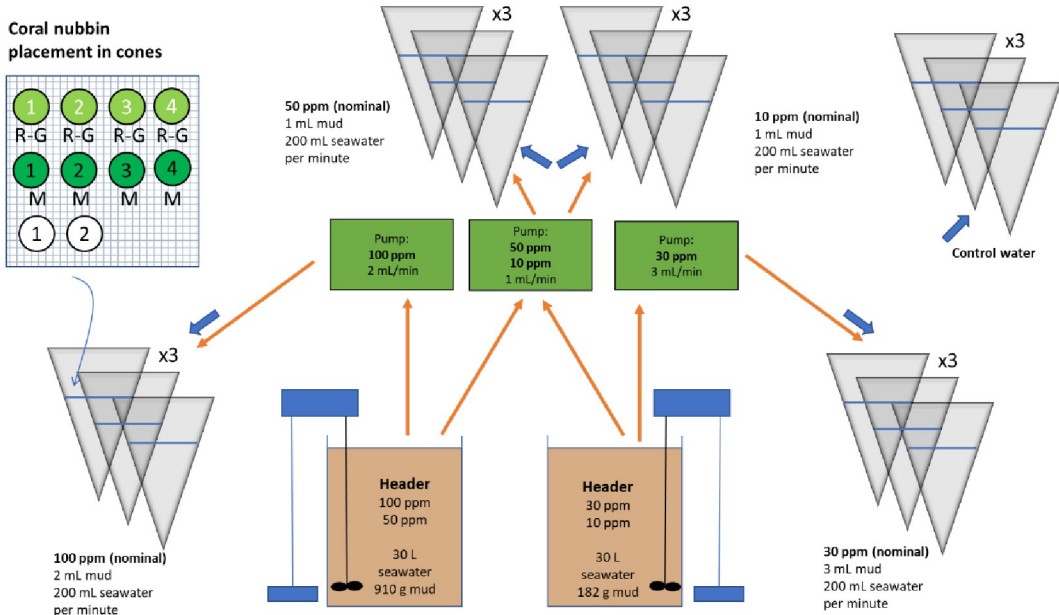

**Fig 1. Design of DC experiment and placement of coral nubbins in replicate cones.** R-G: nubbin used for respiration, growth rate and polyp mortality. M: nubbin used for mucus production. One to two extra nubbins used as backup. The same design principle was used for the BABE experiment but only 3 coral nubbins were used for R-G and M.

in the same cones. A few extra nubbins (1 to 2) were also added in some cones of each treatment for backup. Although it was attempted to use coral fragments with similar morphometry and distribution with regard to polyp counts and sizes in each container and between the two experiments, this was practically challenging given the limitation in coral colonies available for this work and the number of replicate branches needed for each experiment.

The corals were maintained in the same conditions of flow and food in the two experiments. Temperature and salinity were recorded continuously during the experiments (CTD model AquaTroll 100, In-Situ Inc, USA; Temperature accuracy ± 0.1˚C and resolution ~ 0.01˚C; Salinity accuracy ±0.5–1% of reading + 1 mS/cm when reading < 80,000 mS/cm and resolution: 0.1 mS/cm). Both temperature and salinity were maintained as during the acclimation period (7.5 ± 0.2˚C, 33 ± 0.5 PSU). A solution of *A. salina* nauplii was supplied continuously for 8h to each cone using a peristaltic pump (1 ml.min$^{-1}$) three days a week at a mean density of 108±70 and 164±35 *A. salina* nauplii.l$^{-1}$, respectively in DC and BABE experiments. This was equivalent to a total dose of 121 (DC) to 184 (BABE) µmol C.container$^{-1}$. feeding day$^{-1}$ (based on 0.905 µg C -*A. salina* nauplii $^{-1}$, [35]).

## Exposure of corals to suspended particles

Suspended particle (DC, BABE) exposures were prepared as reported in Baussant et al. [28]. Every day, particles were added in pulses of 4 hours followed by 4 hours with no particle addition over a total duration of 5 days. A peristaltic pump added particles at a flow from 1 to 3 ml. min$^{-1}$ into the experimental cones from 30-L stock tanks, where a high-energy mixing kept the material in suspension. The supply of particles mixed with the waterflow was supplied at 200 ±30 ml.min$^{-1}$ to result in target peak exposure concentrations of 10, 30, 50 and 100 mg.L$^{-1}$ for the DC exposure. For BABE, the concentration at 10 mg.L$^{-1}$ was not used.

The exposure period was followed by a 2 weeks recovery in clean seawater when a sampling (T1) was performed. Recovery was thereafter continued for 4 weeks after T1. The experiment was stopped after 6 weeks recovery when a final sampling was performed (T2).

Peak exposure concentrations were controlled from point seawater samples (250mL to 1L) collected each day from the cones during the 4-hour exposure cycle 1 to 2 hours before next DC pump stop. The samples were passed through a GF/F Whatman filter (0.7 µm), and the filter was dried (60˚C) overnight or until constant weight to obtain the total particle weight from which the particle concentrations were calculated [28, 29].

No fractionation of the DC material for particle size was made prior to its use as this was not done in previous studies [28]. However, only the finer material entered the cones and particles coarser than 1 mm either remained in the DC stock or sedimented immediately at the bottom of the cone. Both DC and BABE particles were analysed for grain size and metal composition by an external accredited laboratory (ALS Laboratory Group Norway, Oslo, Norway). A turbidity sensor (model 4112, Xylem Analytics/Aanderaa, Norway) was placed in one replicate cone of each treatment in the DC experiment and the two highest concentrations in the BABE experiment, logging every 5min. the backscattered light from particles in suspension. Turbidity was used here as a qualitative mean to monitor the succession of pulses throughout the exposure but not for actual quantification of the particles. No calibration between particle concentration and sensor turbidity was attempted.

## Coral physiological measurements

Following acclimatization, corals were first sampled for buoyancy weight, respiration, and mucus release to obtain baseline T0 measurements before the start of the exposure. These measurements were used to calculate skeleton growth rate, respiration rate, and mucus production,

respectively, as reported in Baussant et al. [28, 34]. Coral skeleton dry weight (cDW) was estimated using the buoyant weight technique [36] (mean density of *D. pertusum* skeleton used as in [34]) and % skeleton growth rate (%.day$^{-1}$) was calculated from cDW differences between T0 and T1, and between T1 and T2. Respiration rate (mgO$_2$.h$^{-1}$) was normalized to cDW [27, 34, 37, 38]. Mucus production was estimated from POM (particulate organic material) released by the nubbins over 1-day, including particulate organic carbon (OC). OC release was significantly correlated to coral cDW (Linear regression; F:6.608; p = 0.011; r$^2$:0.053) and hence normalised to cDW. Particulate organic nitrogen (ON) was also measured and the OC:ON ratio was also used to evaluate metabolic and mucus quality changes resulting from the exposure of particles. Respiration and mucus measurements were made in static condition with no food present.

The same coral nubbins were repeatedly measured for respiration rate, growth rate, and POM production.

## Determination of polyp mortality

Polyp mortality was measured on the corals used for respiration/growth by visual inspection of each polyp for each coral nubbin both in treatment and control cones. At the start of the experiment, all nubbins were counted and inspected for living polyps. Empty calices were not accounted for. Then, all subsequent determinations for dead and living polyps were made from visual top view observations of the corals maintained underwater to avoid stress from manipulation. Polyp mortality was determined at T1 and T2 by visual re-inspection of the polyps of each nubbin in each cone. A polyp with tissue or tentacles partly retracted in the calyx but still visible was scored as alive. During inspection, the number of dead polyps was counted from which the percentage (%) of dead polyps.nubbin$^{-1}$ was deducted resulting in 3 (BABE) to 4 (DC) measurements per cone. The polyps were considered dead when the tentacles or polyp tissue were consistently retracted inside the calyx, there was no visible tissue (empty calyces), or the calyx was fully smothered with particles. As it can be challenging to be sure of the status of a polyp retracted in the calyx, polyp mortality was determined both during days of feeding and non-feeding to test that polyp retraction was not the result of the absence of *Artemia* feed in the water.

## Data treatment and statistical analysis

Boxplots were employed to summarize graphically the dataset (median, quartiles and outliers/ extreme values) and the data distribution (box = interquartile range, IQR, which contains the middle 50% of the records, and whisker lines that extend from the upper and lower edge of the box) for each coral parameter over experimental time. In such graphical representation, outliers are defined as values between 1.5 and 3 times the IQR, i.e. beyond the whiskers. Extremes values are values more than 3 times the IQR. Outliers and extreme values were not removed from the overall data analysis, taking into consideration that coral nubbin standardization for use in the different cones was challenging and that it was uncertain whether outliers were true incorrectly measured data or a true expression of the large intra- and inter-individual response variability that corals nubbins can show (see [34, 39]). Also, with relatively few coral nubbin replication (n = 3 to 4 nubbins per cone), removal of outlier values would reduce the number of observations and weaken the statistical validity of the analysis. Prior to analysis, the nested factor "replicate tank" was tested for equal distribution of coral measurement variance across the replicate tanks and, given the few number of coral nubbins in each tank (n = 4 to 6), a non-parametric analysis was used. We found no significant difference in coral measurement variance across the different cones (Kruskal-Wallis rank test, p<0.05) and coral data from the

different replicate tanks (n = 3) of treatment and control were pooled. Then a Kolmogorov-Smirnov was applied to test for Gaussian distribution and the assumption of equal variance (homoscedasticity) for the different treatment groups was verified (Levene's test) for $p > 0.05$. These assumptions were met for all measurements but for coral growth rate and polyp mortality for which a Kruskal–Wallis rank non-parametric test was applied, tested for each sampling time i.e. for T1 and T2 ($p < 0.05$) and a Mann-Whitney 2-sample test used as post-hoc analysis. For all other measurements, a repeated ANOVA was applied to test mean differences with treatment concentration as a between-subjects factor and sampling time as a within-subject factor. When the assumption of sphericity was not met (Mauchly's test, $p < 0.05$), p-values were adjusted for repeated time measurements using Greenhouse–Geisser epsilon and Hunyh–Feldt epsilon. A post hoc analysis with Tukey's honest significant difference (Tukey's HSD) test was applied to identify differences between means in treatment and control groups. Polyp mortality scoring determined during the days with and without feeding were pooled if no significant difference was observed or otherwise only days with feeding were used if there was a significant difference (Wilcoxon signed rank test, $p \leq 0.5$).

All statistical analyses were performed with the software package SPSS© (v. 25.0, IBM, New York, NY, USA).

## Results and discussion

### Analysis of particle size distribution and metal content of DC and BABE

S1 Appendix shows the DC, BA and BE composition (Fig A and B in S1 Appendix) and particle distribution (Fig C in S1 Appendix) performed by the ALS laboratory. The main difference between the different materials was the high silica content of DC and BE whilst BA was characterized by a high content of barium sulphate and strontium. Further BE had a much finer particle size distribution, with $>95\%$ with a size $< 10$ μm, compared to BA and DC (50% with a size $< 10$ μm). For BA, however, no particle exceeded a size of 100 μm whilst for DC, ~30% of the particles were $>100$ μm.

### Experimental concentration in both experiments

Turbidity measures performed continuously in cones of each treatment showed a repeated peak exposure of DC (S2 Appendix) and BABE (S3 Appendix) 3 times a day over the 5 days of exposure and a gradient of exposure concentration as anticipated from exposure scenario and target concentrations. Turbidity increased rapidly at the onset of each exposure pulse and reached peak exposure within ~2hours, then decreased gradually again to baseline level between two exposure pulses. Turbidity measurements indicated that peak actual particle concentration in water (1FTU ~ 1 mg/l) did not reach the DC target concentrations (10, 30, 50 and 100 mg.l⁻¹) nor the BE target concentrations (50 and 100 mg.l⁻¹). For BA, turbidity measurement showed suspended particles concentration that was twofold higher the target concentrations (50 and 100 mg.l⁻¹), whilst it was twofold lower based on filter measurements. Possibly barite particles reflect light in a different manner to the other particles, necessitating a calibration curve to reflect absolute values. Turbidity was used qualitatively in this study to visualize and monitor the exposure pulses, not as a proxy for quantitative measurement of actual suspended particle concentrations, which was relying on filter dry weight measurement.

Overall, based on weight measurements from water filtration, we observed ~ 50% decrease in the actual suspended particle concentrations compared to the nominal target concentrations (Table 1). Over the entire DC exposure, the mean actual peak DC concentrations were ~4, 14, 19 and 49 mg.l⁻¹, respectively corresponding to DC target concentrations of 10, 30, 50 and 100 mg.l⁻¹.

**Table 1. Target concentrations are nominal concentrations expected in this study.** Measured concentrations are the mean±standard deviation (stdev) of actual drill cutting (DC), barite (BA) and bentonite (BE) particles concentration (mg.l$^{-1}$) in the cones in both studies estimated from filters. Actual concentration are the rounded values for DC, BA and BE mean measured concentration, further used in this article to describe the exposure.

| Experiment | Nominal mg.l$^{-1}$ (target) | Measured mg.l$^{-1}$ | | Actual mg.l$^{-1}$ (rounded) |
|---|---|---|---|---|
| | | Mean | Stdev | |
| Field DC | 10 | 4.4 | 0.6 | 4 |
| | 30 | 13.6 | 3.8 | 14 |
| | 50 | 18.7 | 4.9 | 19 |
| | 100 | 49.2 | 5.7 | 49 |
| BA particles | 30 | 12.2 | 1.6 | 12 |
| | 50 | 25.5 | 2.9 | 26 |
| | 100 | 63.1 | 4.3 | 63 |
| BE particles | 30 | 10.8 | 0.6 | 11 |
| | 50 | 22.6 | 1.3 | 23 |
| | 100 | 48.0 | 3.9 | 48 |

For BA, actual measured peak concentrations were 12, 26 and 63 mg.l$^{-1}$ (corresponding to respectively target concentration of 30, 50 and 100 mg.l$^{-1}$) and for BE, this was 11, 23 and 48 mg.l$^{-1}$ (corresponding to respectively 30, 50 and 100 mg.l$^{-1}$) (Table 1).

Also, Järnegren et al. [31] found an average decline of ~50% for DC, BA and BE particles in their experimental setup with gentle agitation of their glass vial containers over 24h exposure time. Clearly, even the relatively fine drilling material (<100 μm) can rapidly sink in the water with low agitation but, in the field, this might be otherwise. For example, Frost et al. [15] report average current speed of 6–7 cm/s near the seabed with maximum current speed of the order of 12–16 cm/s at about 250 m depth at a drilling site in the Norwegian Sea. However, because of their size distribution, BA and particularly BE are expected to spread in the water column rather than sink to the bottom. For DC particles, sinking velocities are expected to be higher [40]. Overall, the concentration measured in the cones are in the order of magnitude of what is measured in the field on average in form of spikes above seabed during a drilling period [15, 41].

## Effects on coral physiology and polyp mortality

A summary of the observations made in this study is reported in Table 2 based on statistical significance.

**Table 2. Summary of the effects on *D. pertusum* physiology and polyp mortality in the DC, BA and BE experiments.** 0 no significant effect 1 significant (p≤0.05) effect.

| Experiment | Exposure | Respiration rate | Growth rate | OC release | OC:ON | Polyp mortality |
|---|---|---|---|---|---|---|
| **DC** | 4 | 0 | 0 | 0 | 0 | 0 |
| | 14 | 0 | 0 | 0 | 0 | 0 |
| | 19 | 0 | 0 | 0 | 0 | **1** |
| | 49 | 0 | 0 | 0 | 0 | **1** |
| **BA** | 12 | 0 | 0 | 0 | 0 | 0 |
| | 26 | 0 | 0 | 0 | 0 | 0 |
| | 63 | 0 | 0 | 0 | 0 | 0 |
| **BE** | 11 | 0 | 0 | 0 | 0 | 0 |
| | 23 | 0 | 0 | 0 | **1** | 0 |
| | 48 | 0 | 0 | 0 | **1** | 0 |

A detailed analysis and discussion of these results follows below.

**Respiration rate.** Respiration rate reflects the metabolic activity to support basic functions of CWC, which is influenced by both feeding and temperature [42, 43]. Under high feeding regime, respiration increases as a response to the higher metabolic rate to process food, while a decrease in respiration is observed subsequent to sub-optimal feeding or starvation conditions [32, 34, 44]. Variability in temperature is also known to influence significantly $O_2$ consumption [43, 45].

Here, the mean respiration rate prior to exposure of DC and BABE was as reported previously by Baussant et al. [34] in a similar experimental setup with similar food regime (i.e. 0.0038±0.002 mg O2/h/g cDW, Fig 2). Temperature was also maintained constant and stable throughout these experiments, possibly explaining low variability in $O_2$ consumption. There was no influence of time nor exposure to DC or BABE material in coral respiration rate. Under exposure to solid particles in the water, one could expect an increase in respiration rate consequent to the higher metabolic activity resulting from the removal of particles by polyps and mucus secretion. However, other laboratory studies, even with longer exposure to DC particles in the range 2~50 mg.l$^{-1}$ for 2.5 (continuous) to 12 (discontinuous) weeks, show no significant change in respiration rate [28, 29]. It seems respiration is very resilient to change from particle exposure in the laboratory. It has been suggested that this might be due to optimal coral feeding in laboratory experiments, masking the real effect of stressors on the respiration rates [42]. To our knowledge, no study has looked at the effect on respiration rate to drilling or sediment particles on starved CWC. *D. pertusum* seems to cope relatively well with long periods of food deprivation, still able to grow but respiration is reduced whilst overfed corals seem to increase respiration [32, 34]. It could be interesting in future investigations to measure effects on coral respiration from drilling suspended particles under sub-optimal food condition to eliminate the masking effect of feeding.

**Growth rate.** Compared to tropical corals, CWC grow slowly, and compared to Mediterranean conditions, the rate of calcification in *D. pertusum* of Atlantic water is even lower [42].

The method used here for skeleton growth rate provides a total nubbin weight gain but does not allow to measure the differences in growth from thicker older polyps and thinner younger polyps. Different polyp generations between the nubbins might lead to consequent higher growth rates in some nubbins from some colonies, than others. Generally, there was a relatively high variation in skeleton growth rate between coral nubbins. This is not uncommon and is challenging to harmonize for experimental work with *D. pertusum*. In other studies, it has been shown that young polyps grow faster than old ones [39, 46]. Hence, the variation observed is most likely explained by the differences in polyp age of the coral nubbins used in the experiments. In the selection of nubbins, it was attempted to use those originating from a same colony and distribute them into the different experimental cones to mitigate this effect but it was needed to also use nubbins from other colonies with different polyp size and age, which could explain the variability of growth rate between nubbins of different colonies in the same container.

The mean skeleton growth rate of control coral nubbins in the DC experiment was 0.03% ±0.03%.day$^{-1}$ between T0 and T1 (ΔT1T0) and 0.056%±0.03% from T1 to T2 (ΔT2T1) (Fig 3). These values are very much consistent to previous laboratory measurements for *L. pertusa* [28, 29, 32, 34, 39], field measurements [12] and other CWCs [42]. Compared to the DC experiment, mean skeleton growth rate of control *D. pertusum* nubbins in the BABE experiment was lower (ΔT1T0: 0.009%±0.007%.day$^{-1}$; ΔT2T1: 0.010%±0.009%; whole experimental period:0.0085±0.0055%.day$^{-1}$), however still within the range of what previously laboratory and field investigations have shown [47, 48].

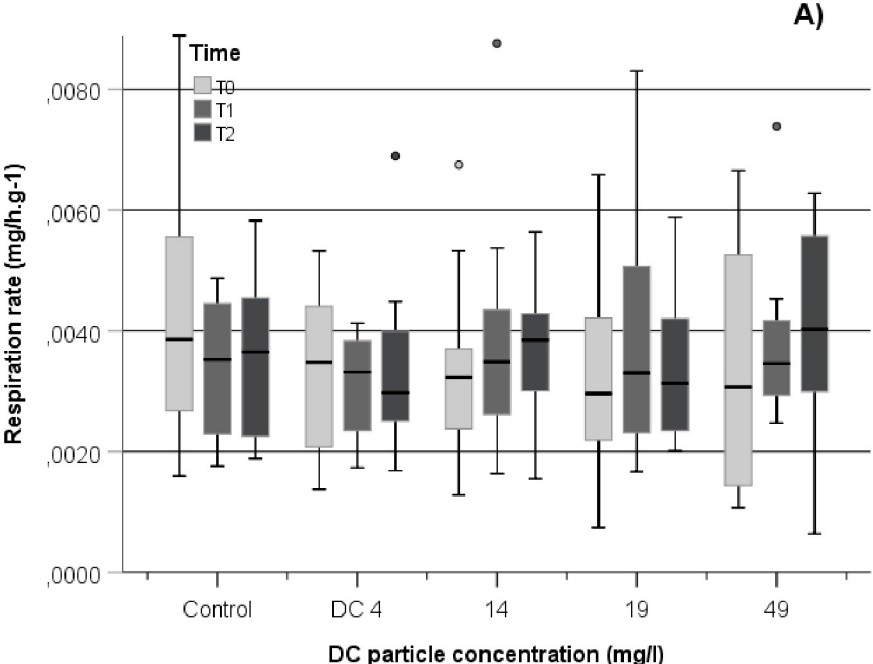

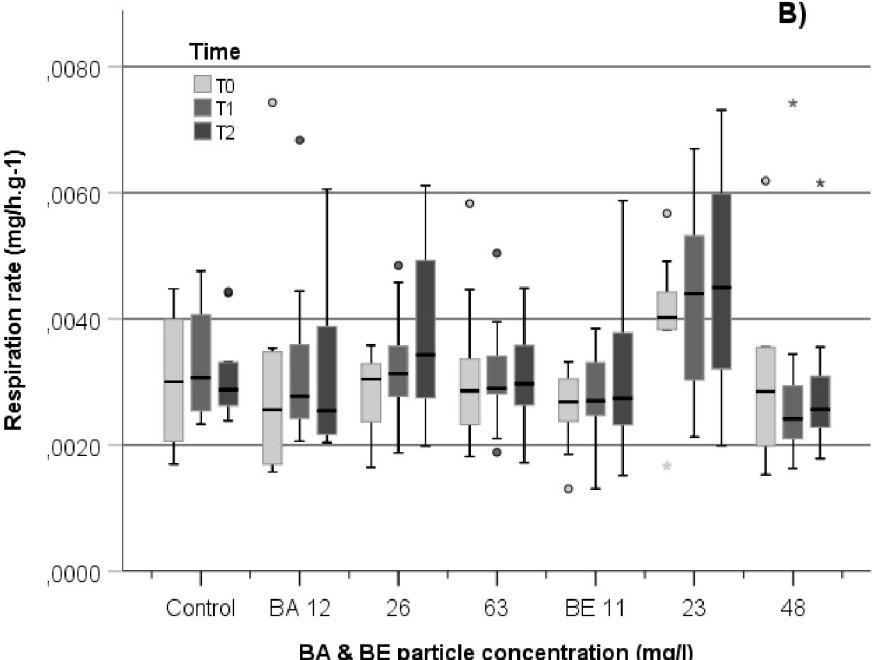

**Fig 2. Boxplot of the respiration rate of *D. pertusum* in the DC experiment (A) and in the BABE experiment (B) measured at T0 (prior to exposure), T1 (post-exposure+2 weeks recovery), and T2 (+4weeks recovery).** The line in the box is the median value. The bottom and top bars in the boxes represent the 25th and 75th percentiles; whiskers above and below the box are the 10th and 90th percentiles, respectively. Outliers are visualized with circles and extreme values with stars, beyond whisker lines. For each treatment and control group, the data from each replicate tank are pooled at each time point both for DC (n = 12) and for BABE (n = 9).

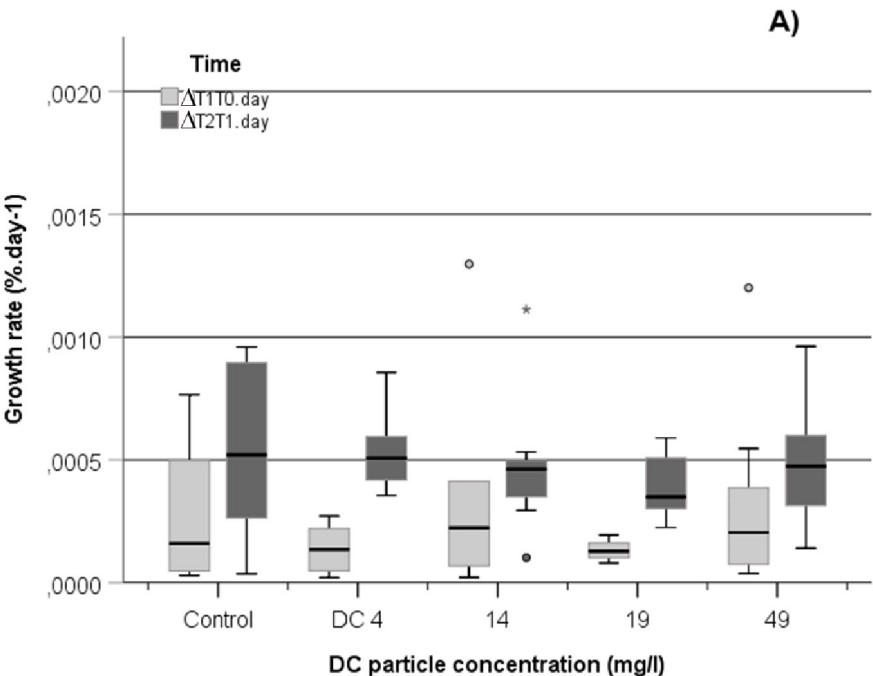

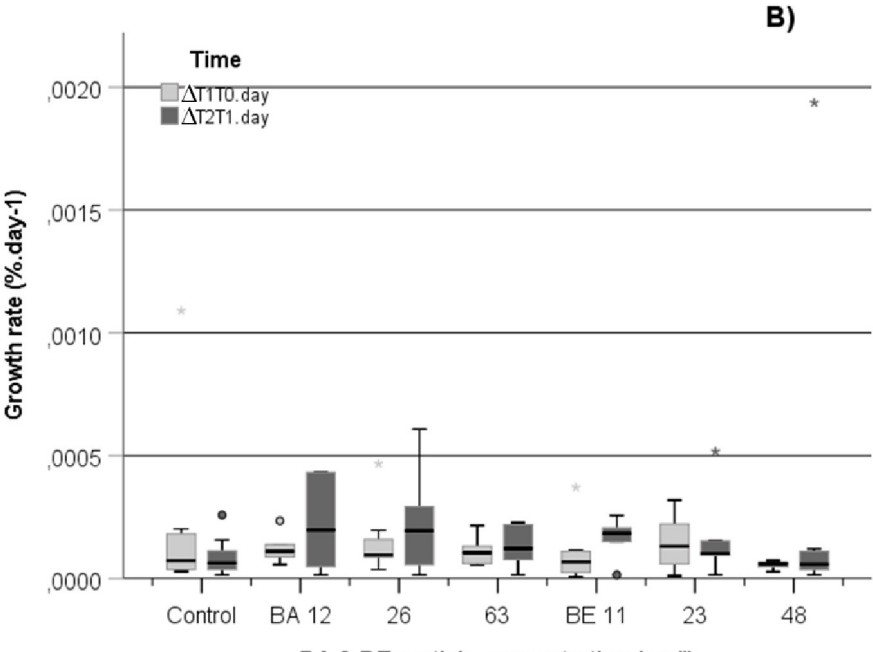

**Fig 3. Boxplot of the skeleton growth rate of *D. pertusum* in the DC experiment (A) and in the BABE experiment (B).** ΔT1T0 is the skeleton growth rate measured from T0 (prior to exposure) to T1 (2 weeks recovery), and ΔT2T1 is the skeleton growth rate measured between T1 an T2 (6 weeks recovery). The line in the box is the median value. The bottom and top bars in the boxes represent the 25th and 75th percentiles; whiskers above and below the box are the 10th and 90th percentiles, respectively. Outliers are visualized with circles and extreme values with stars, beyond whisker lines. For each treatment and control group, the data from each replicate tank are pooled at each time point both for DC (n = 12) and for BABE (n = 9).

Possibly, skeleton growth rate is reduced after seasonal control. The BABE experiment was performed between December and January, when a high percentage of the coral tissue pool and coral energy is diverted to gametes maturation prior to spawning in February/March [32, 49]. In the field, Maier et al. [43] found a distinct seasonality of skeleton growth with an increase from February to late May coinciding with the spring planktonic bloom, but a reduction in December-February coinciding with the spawning season of *L. pertusa*.

Overall, in the DC-experiment, growth rate measured from T0 to T1 (ΔT1T0) was significantly lower than that measured between T1 and T2 (ΔT2T1) (Wilcoxon signed rank test; p = 0.001). Even so, the distribution of growth rate values was equal in control and in DC-exposed corals at all time point. It is possible that the corals grew better from T1 to T2 due to a longer acclimatization to the experimental conditions than before T1. Also, there was no significant difference to control with any of the BABE-exposed corals at any time. Skeleton growth rate measured after T1 was the same as the one measured after T2. Skeleton growth rates measured on the coral nubbins used for mucus production showed the same trends and no difference between treatments in the DC nor in the BABE experiments (S4 Appendix).

After Maier et al. [50], the energy required for calcification is a small fraction (1–3%) of the total metabolic energy demand and corals, even under unfavourable conditions such as low food conditions, might still be able to allocate this small portion of energy to calcification [32, 34]. In the field, Lartaud et al. [51] found a significant difference in growth rate resulting from natural sedimentation and smothering of polyp tissue leading to polyp mortality after 15 months of deployment. In the lab study by Baussant et al. (2018), skeleton growth rate of *D. pertusum* was not significantly changed following 2.5 continuous and 12 weeks discontinuous exposure to DC treatments in a range of concentrations from 2~50 mg.l$^{-1}$. Likewise none of the DC treatments (5 and 25 mg.l$^{-1}$) differed significantly from the control treatment after a period of 12 weeks in the experiment by Larsson et al. [32]. Both Larsson et al. [32] and Baussant et al. [28] reported an increase in skeleton growth rate in presence of low amount of DC/sediment particles (~5 mg.l$^{-1}$), compared to high particle loading, possibly resulting from the direct (particles associated to DC) or indirect (microbes-bound to particles associated to DC) ingestion of organic matter from DC. However, in this study, the apparent positive effect of DC on skeleton growth rate was not observed, possibly due to the shorter exposure time.

**POM production.** Like tropical corals, CWCs produce and release a substantial amount of organic matter into their environment as dissolved (DOM) and particulate matter (POM) [44, 52], including organic carbon (OC) and organic nitrogen (ON), which is used for different processes such as feeding or cleaning coral surfaces [52–55].

The proportion of DOM or POM release by corals seems to vary according to food density [34], water current condition [56], or other environmental stress such as under the presence of suspended particles [28, 29, 55]. The release of organic matter by corals can represent a significant component of their energy budget [29, 44] whilst a higher turnover due to stressors such as sedimentation is expected to have a negative consequences for their energy balance or the derivation of this energy to other basic processes than growth and reproduction [44]. There is a complex and ecologically important turnover of mucus-related organic matter which either can be re-ingested by the corals [57] or used to fuel other reef inhabitants [58] and microbial pathways recycling mucus-derived organic material [59]. Hence mucus production is not only an important function of the coral physiology but as well for the whole coral ecosystem by providing a significant pool of organic matter [57].

Under stress from suspended particles, the coral was expected to produce more mucus and the content of OC in the incubation water to increase. Baussant et al. [28] found a significant difference in OC production following 12 weeks of discontinuous exposure to a mean concentration of 25 mg.l$^{-1}$ (peak exposure of 50 mg.l$^{-1}$) suspended DC.

Compared to basal OC in seawater only, OC from water incubated with all coral nubbins increased significantly, indicating a substantial release of coral mucus-related OC during the incubation period (Fig 4). OC release increased between the two sampling events with BE exposure (repeated ANOVA; p = 0.061) but the combination of time and concentration was not significantly changed in any treatment compared to control. Also, the release of OC by corals was not significant different to the control coral nubbins at any sampling time in DC and BA exposure.

The OC:ON ratio of mucus POM was significantly higher in seawater only (>15) compared to the water incubated with corals (~8–9) in both experiments (Fig 5). OC:ON ratios of 5 to 7 have been measured for *Lophelia pertusa* [58, 60]. There was no change with sampling time nor effect in the mucus OC:ON ratio with any DC treatment, however OC:ON decreased significantly with time in BA treatment (repeated ANOVA; p = 0.032) but this observation was not different in BA-treated and control coral nubbins. In the BE experiment, there was an increase of OC:ON with time and concentrations (repeated ANOVA, time x concentration: p = 0.054) in corals exposed to elevated levels (23 and 48 mg.l$^{-1}$). Larsson et al. [29] also found that total OC increased between water only and water with corals but their study did not show any significant change in OC production when exposed to suspended barite (25 mg.l$^{-1}$). The interpretation of the OC:ON difference in *D. pertusum* exposed to high BE particles is uncertain. OC production increased between T0 and T1 in BE treatments, it is possible that the corals retained or reabsorbed relatively more of the protein-rich ON fraction in their tissue, explaining the increase in the mucus OC:ON ratio at high BE particle concentration. BE is a finer solid than BA and DC and may have been incorporated to a greater proportion in coral tissues. Reabsorption of coral mucus has been found in situation where corals are stressed such as under low food regime [61] and corals seem to have developed ways to regulate mucus production and composition to reduce energy loss when they need to [44]. However, no tissue measurement of OC and ON content was added to this study, so we cannot verify this with the present data.

**Polyp mortality.** In the DC experiment, we found that polyp mortality determination measured during days with addition of food particles and days without food addition was equal (Wilcoxon signed rank test, p = 0.103). Hence, data from days of feeding and no feeding were pooled to test differences in polyp mortality resulting from DC treatments at T1 and T2. However, in the BABE experiment, there was a statistically significant difference in polyp mortality scored during days with feeding and days without feeding (Wilcoxon signed rank test, p = 0.032 and 0.002 respectively for BA and BE exposure). Hence the T1 and T2 for the BABE dataset were based on polyp mortality for days with feeding only.

In our experiment, polyp mortality was variable between coral nubbins in all treatments both in DC and BABE experiments (Figs 6 and 7). Overall, for both experiment, polyp mortality in control coral was ~20%, which was attributable to one or two of the three replicates missing some of their polyps with no clear reason. The maintenance of corals in the laboratory after collection of the coral colonies, coral handling to prepare the nubbins, or the environmental conditions in the experimental cone may be possible explanations. However, corals used in the BABE experiment were maintained longer in the laboratory before experimental start than those used in the DC experiment, but polyp mortality was identical in both experiments. Coral handling could provide a cause for coral mortality as the same corals nubbins were used for respiration and growth rate measurements, which implied some transfer of the corals for incubation.

Nevertheless, in DC experiment, polyp mortality was significantly higher (Kruskal-Wallis test, p = 0.037) to control at 19 and 49 mg.l$^{-1}$ at T1 sampling (*Post Hoc* pairwise comparisons). Polyp mortality at T2 also showed the same trend with an increase at high DC concentrations

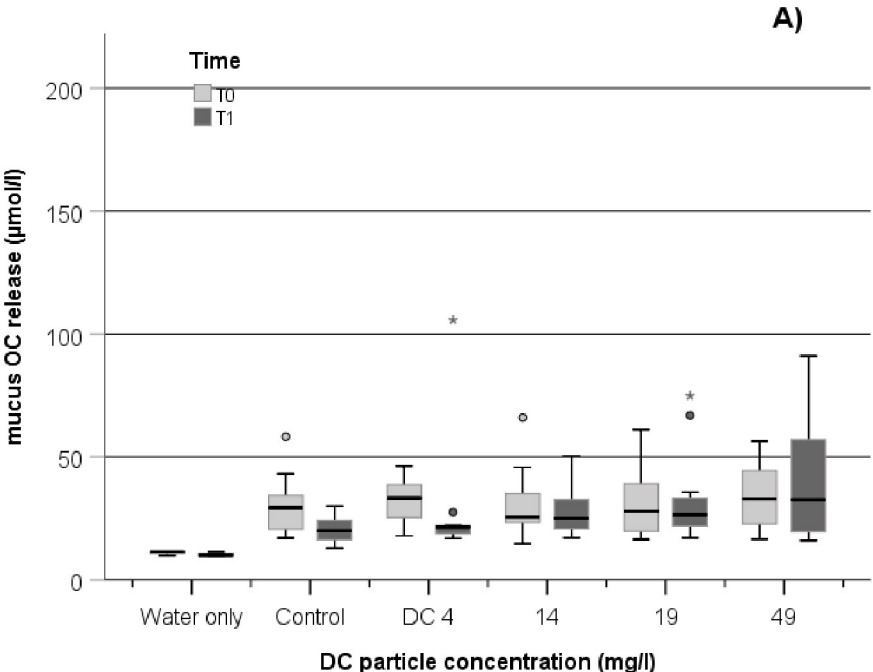

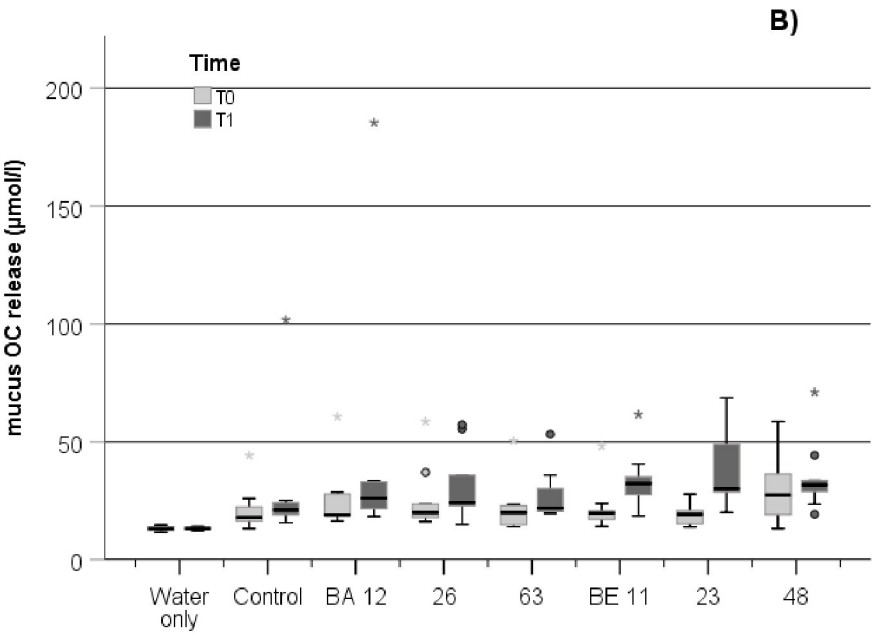

**Fig 4. Boxplot of the OC mucus release by *D. pertusum* in the DC experiment (A) and in the BABE experiment (B) measured at T0 (prior to exposure) and T1 (post-exposure+2 weeks recovery).** The line in the box is the median value. The bottom and top bars in the boxes represent the 25th and 75th percentiles; whiskers above and below the box are the 10th and 90th percentiles, respectively. Outliers are visualized with circles and extreme values with stars, beyond whisker lines. For each treatment and control group, the data from each replicate tank are pooled at each time point both for DC (n = 12) and for BABE (n = 9).

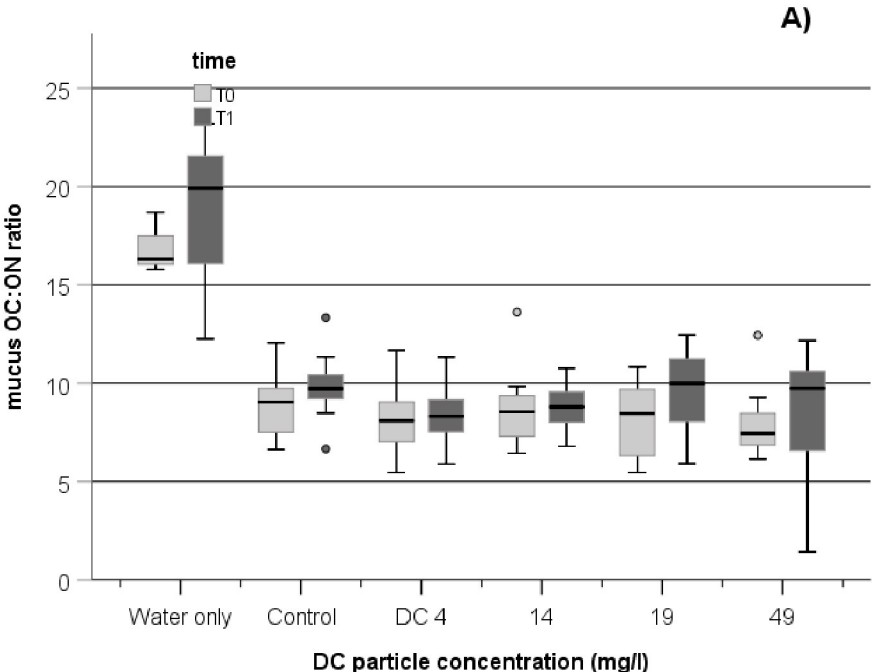

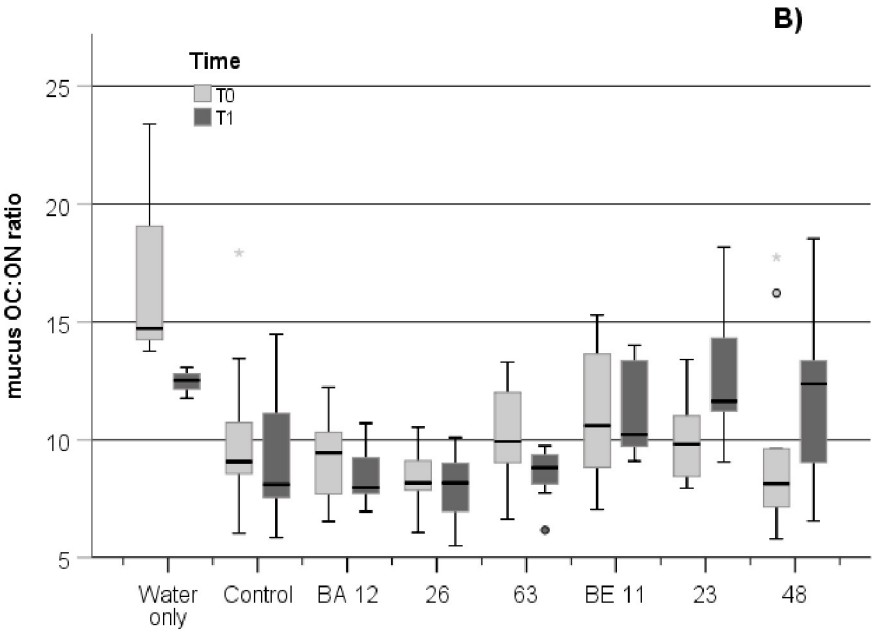

**Fig 5. Boxplot of the mucus OC:ON ratio in mucus release by *D. pertusum* in the DC experiment (A) and in the BABE experiment (B) measured at T0 (prior to exposure), and T1 (post-exposure+2 weeks recovery).** The line in the box is the median value. The bottom and top bars in the boxes represent the 25th and 75th percentiles; whiskers above and below the box are the 10th and 90th percentiles, respectively. Outliers are visualized with circles and extreme values with stars, beyond whisker lines. For each treatment and control group, the data from each replicate tank are pooled at each time point both for DC (n = 12) and for BABE (n = 9).

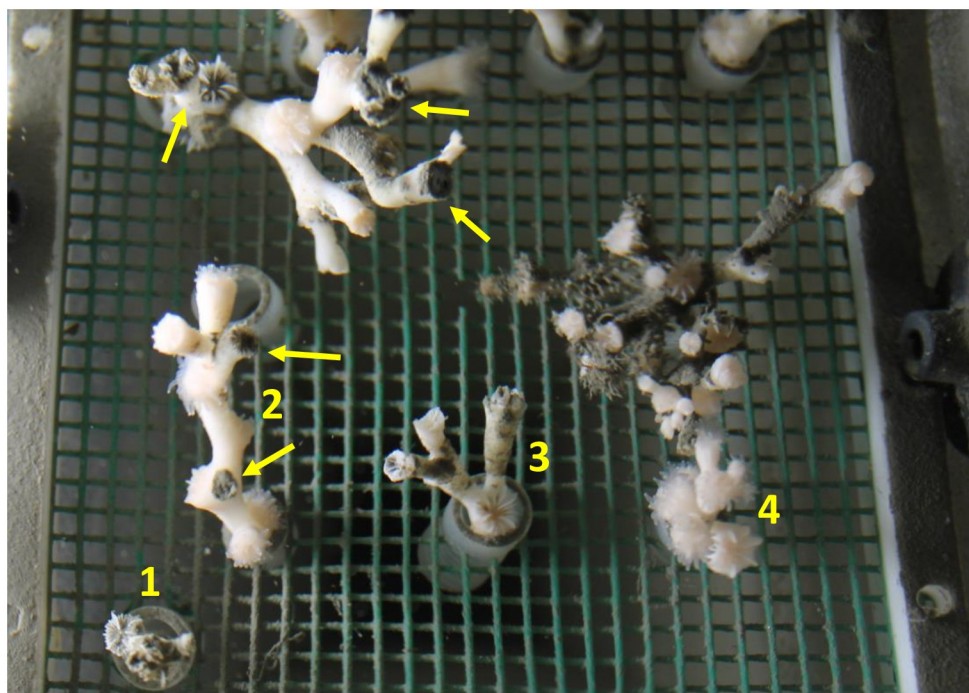

**Fig 6. Top view of the coral nubbins in replicate cone 3 of the DC 49 mg.l$^{-1}$ exposure at T1 sampling.** Nubbins 1–4 are those used for polyp mortality determination. Note large difference in surface DC coverage from one coral to the other. Here the polyps of nubbins 1 and 3 are most affected by DC smothering. Arrows: areas of coral with heavily covered surface and entrapped DC around the polyp calyxes.

(Kruskal-Wallis test, p = 0.083) but not different to polyp mortality at T1. A possible cause of this mortality was an accumulation of the DC material on the surface of corals after the 5 days of exposure. Post-exposure, the DC material remained on surfaces of some of the corals, which could cause smothering and mortality of some polyps (Fig 6). The variability in polyp mortality between coral nubbins within replicate tanks may be explained by the condition and coverage of coenosarc between coral nubbins in the tanks. In the laboratory, it has been shown that the surface of the skeleton and calyx covered with coenosarc always remain clean and with no sedimentation of particles [62], whereas areas with bare skeleton facilitate the settling of particles that can become progressively smothered with drill cutting depositions over time [27], potentially causing death of the polyps.

In the BABE experiment, polyp mortality increased significantly between T1 and T2 (Wilcoxon signed rank test: p = 0.023 and p = 0.001, respectively for BA; and BE exposure). There was a higher occurrence of polyp mortality in both BA and BE-exposed corals at T2 at concentration ≥12 mg.l$^{-1}$ and ≥23mg.l$^{-1}$, respectively. However, the variability between corals was high and the median polyp mortality remained low, with no significant difference to control polyp mortality.

Polyp mortality of corals in the laboratory or *in situ* has been shown to be variable between studies. Büscher et al. [48] observed 10–30% polyp mortality on average in their *in situ* growth experiment in different Norwegian fjord and offshore locations but the same authors found less than 8% polyp mortality on average in a laboratory study mimicking field-like conditions [47]. Larsson et al. [29] found that polyp mortality was very low during their 3 months long exposure period with no polyp mortality observed in their control corals, only 2.2% in their high DC exposure and 0.3% in their high natural sediment treatment both at concentration of

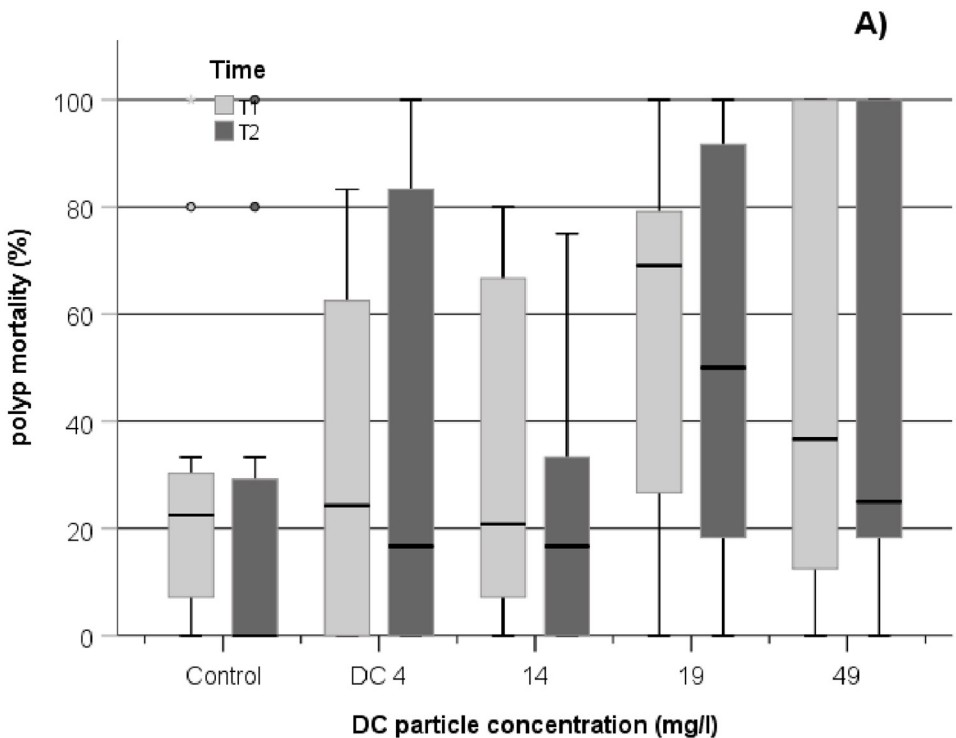

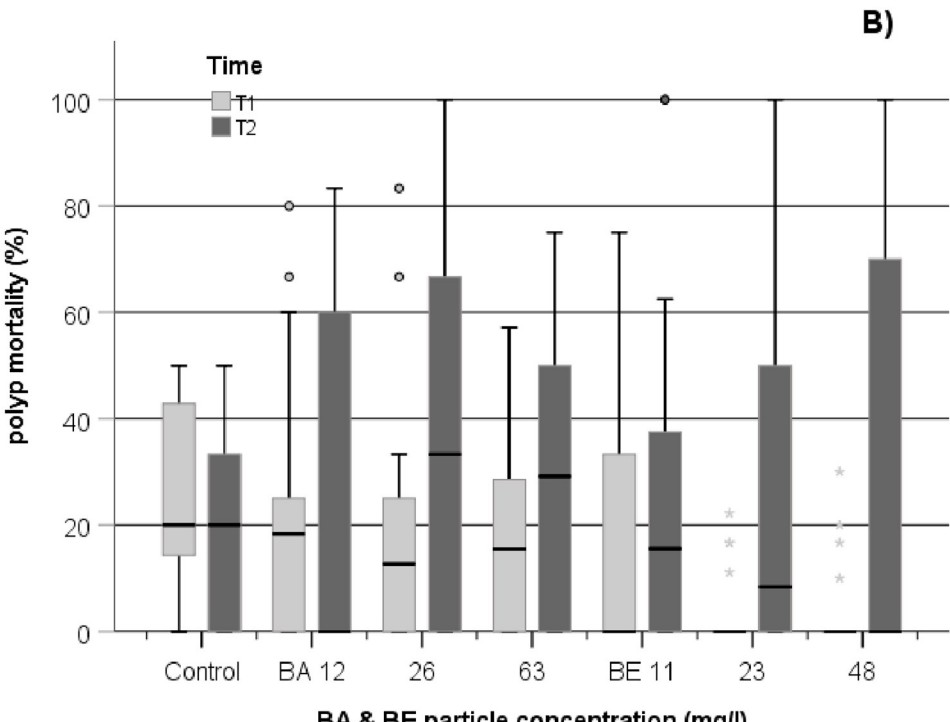

**Fig 7. Boxplot of polyp mortality (% polyps.nubbin[-1]) of *D. pertusum* in the DC experiment (A) and in the BABE experiment (B) measured at T1 (post-exposure+2 weeks recovery), and T2 (+4week recovery).** The line in the box is the median value. The bottom and top bars in the boxes represent the 25th and 75th percentiles; whiskers above and below the box are the 10th and 90th percentiles, respectively. Outliers are visualized with circles and extreme values with stars, beyond whisker lines. For each treatment and control group, the data from each replicate tank are pooled at each time point both for DC (n = 12) and for BABE (n = 9).

25 mg.l$^{-1}$. However a much higher polyp mortality was determined by Brooke et al. [63] who exposed corals to a suspension of sediment concentration ranging from 54 to 362 mg.l$^{-1}$ for 14 days. They found that ~10% of polyp died at 54 mg.l$^{-1}$, and >40%, > 60% and >90% at 103, 245 and 362 mg.l$^{-1}$, respectively.

## Conclusions and outlook

In the deep-sea, CWCs are key formations, fuelling a diverse and productive ecosystem. In the recent decades, CWC habitats have seen severe degradation mostly from bottom trawling, and with a changing ocean, CWC may experience further loss from an increasing combination of stressors [3]. Both resilience and response to these stressors are uncertain, and there is a lack of well-defined tolerance levels for individual as well as combined stressors. This research contributes to support in coral risk assessment related to offshore petroleum, providing managers effect thresholds to coral endpoints from realistic scenarios of exposure with drill particles in suspension. These thresholds are needed by the petroleum industries for evaluating the environmental risk and mitigation actions to consider for current drill sites co-occurring with coral formations but as well for new exploration. Here, the work mimicked short 5-days exposure to repeated pulses of drill particles, which represents a more realistic exposure to corals in the field compared to long (weeks) and continuous exposure used in former laboratory studies with CWC such as *D. pertusum*. Generally, our observations confirmed those reported previously by other studies that the coral *D. pertusum* (previously *L. pertusa*) is relatively resilient to effects from exposure mimicking different scenarios of drill particles [26–29]. They indicate that coral mucus is a relatively good marker of coral exposure although a standardization of the collection of mucus is warranted and the mucus end-points (qualitative or quantitative, [55]) to measure should be well identified and standardized too. The significance of the change observed in mucus OC:ON in the BE experiment (23 and 48 mg.l$^{-1}$) for coral function, however, needs further insights. Although exposure was short and in pulses, polyp survival was significantly affected by high DC concentrations (19 and 49 mg.l$^{-1}$) but polyp mortality was generally high, also in control treatment. At the onset of this study, we assumed that short-term exposure to suspended drill particles would increase the tolerance threshold of the coral *D. pertusum* that was previously estimated in long-term exposure to DC at ~10 mg.l$^{-1}$ [28]. Overall, we found a threshold of effects to the suspended drill material used herein at a level of ~20 mg.$^{-1}$, where a significant increase in mucus OC:ON (BE experiment) and a significant increase in polyp mortality (DC experiment) were observed. Nonetheless, we have only indication that the limit value for BA is at the same level (~ 20 mg.l$^{-1}$) through this study and further confirmation of this is needed. Also, compared to the study of Järnegren et al. [31] using 8- and 21-day old *D. pertusum* coral larvae and other coral studies with different sediment types [24, 64], the adult stage of the coral did not appear more affected to suspended BE than suspended DC nor BA particles and, although only few, different endpoints were affected 2 and 6 weeks after exposure ceased. Hence, from this research, it appears that for assessing the environmental risk to coral colony of *D. pertusum* to suspended drill particles, the particle load rather than the particle type matters.

Whilst corals in this and previous studies appear relatively resilient to long-term effects from drill particle exposure, there are some uncertainties, which should be looked upon in future research. Corals obtained from other biogeographical regions or with other genetic and environmental differences might respond differently [39]. There are also intrinsic discrepancies in laboratory experiments compared to the natural environment, with regards to flow, trophic conditions, and occurrence of distinct coral morphotypes. Here, we used corals collected in fjord systems and they may be more accustomed to a higher variability in environmental

conditions, such as from tidal events, and tolerate larger changes in their environment, than offshore corals. Coral morph differences have been suggested to influence resilience to environmental changes such as from ocean acidification, with the higher pigmented orange corals showing broader range of net growth rates, as well as significantly lower polyp mortality than white specimens [48]. However, Brooke et al. [63] found no difference in suspended sediment tolerance using two distinct growth forms. Provan et al. [65] showed coral morph differences in protein patterns in the mucus of coral maintained in the laboratory, possibly indicating differences in functional and metabolic pathways to cope with their environment. Hence, other trends in effects measured for mucus OC:ON, polyp mortality and other parameters could have been observed with corals from other biogeographical regions or the use of distinct coral nubbins morphotypes (color, growth form).

Today, operators rely on risk models to identify "mitigation responses" where drilling discharge plan needs to be revised with additional technology or modified to avoid impacting on potentially sensitive communities. However, there is still a poor understanding of the relationship between suspended particle exposure and the response of corals and we still lack a meaningful link between the change in coral end-point measurements in the laboratory and their actual consequences for coral (except for polyp mortality such as reported in this study, but which is shown to vary largely between corals). The present research is important to support and refine these models with data obtained from realistic exposure for better predictions and to avoid damages to coral formation from offshore discharges as part of industry's licence to operate. Considering the variability in discharge scenarios, the above-mentioned uncertainties and variability of responses on coral, we hope this research can be followed up, taking into considerations some of these variables, to provide more secure and ecologically meaningful thresholds for risk model. As ocean is also changing, so will the risk to coral change [3, 66]. Hence the combination of realistic offshore discharge scenarios and other upcoming anthropogenic impacts [67] is warranted in future coral experimental work for a comprehensive prediction of the consequences on these unique hotspots of life and adjust management measures from petroleum discharge consequently [68].

## Supporting information

**S1 Appendix. Composition (A,B) and particle size distribution (C) of DC, BA and BE material using in the experimental studies.**
(DOCX)

**S2 Appendix. DC turbidity measurements at the different DC target exposures during the experimental period.** 1FTU ~ 1 mg.l⁻¹. Data recorded every 5 min.
(DOCX)

**S3 Appendix. BA and BE turbidity measurements at the 50 and 100 ppm target exposures during the experimental period.** 1FTU ~ 1 mg.l⁻¹. Data recorded every 5 min.
(DOCX)

**S4 Appendix. Boxplot of the skeleton growth rate of *D. pertusum* nubbins used for mucus production in the DC experiment (A) and in the BABE experiment (B).** DT1T0 is the skeleton growth rate measured from T0 (prior to exposure) to T1 (2 weeks recovery), and DT2T1 is the skeleton growth rate measured between T1 an T2 (6 weeks recovery). Nominal concentrations are used on *X*-axis. The line in the box is the median value, and whiskers are the highest and lowest data below 1.5xIQR. Outliers are visualized with circles and extreme values with stars.
(DOCX)

## Acknowledgments

We are particularly grateful to Tone K. Frost and Ingunn Nilsen at Equinor AS for support, advices in the design of the study, discussions of the results, decision to publish and several comments to improve the manuscript. The authors also wish to thank the captain and crew of R/V Gunnerus for collection of coral samples and Johanna Järnegren at the Norwegian Institute for Nature Research (NINA) for a precious help and advice regarding collection.

## Author Contributions

**Conceptualization:** Thierry Baussant, Ingrid Myrnes Hansen.

**Data curation:** Thierry Baussant, Maj Arnberg, Emily Lyng, Sreerekha Ramanand, Ingrid Myrnes Hansen, Dick Van Oevelen, Peter Van Breugel.

**Formal analysis:** Thierry Baussant, Maj Arnberg, Emily Lyng, Sreerekha Ramanand, Dick Van Oevelen, Peter Van Breugel.

**Funding acquisition:** Thierry Baussant, Ingrid Myrnes Hansen.

**Investigation:** Thierry Baussant, Maj Arnberg, Emily Lyng, Sreerekha Ramanand, Shaw Bamber, Mark Berry.

**Methodology:** Thierry Baussant, Maj Arnberg, Emily Lyng, Sreerekha Ramanand, Shaw Bamber, Mark Berry, Ingrid Myrnes Hansen, Dick Van Oevelen, Peter Van Breugel.

**Project administration:** Thierry Baussant.

**Resources:** Thierry Baussant, Dick Van Oevelen, Peter Van Breugel.

**Supervision:** Thierry Baussant.

**Validation:** Thierry Baussant, Maj Arnberg, Dick Van Oevelen, Peter Van Breugel.

**Visualization:** Thierry Baussant, Maj Arnberg, Emily Lyng, Sreerekha Ramanand, Ingrid Myrnes Hansen.

**Writing – original draft:** Thierry Baussant, Maj Arnberg, Ingrid Myrnes Hansen.

**Writing – review & editing:** Thierry Baussant.

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
