## [Decision Letter · Decision Letter 0]

27 Oct 2021

PONE-D-21-17662Cover Letter for maunuscript with the title “Identification of tolerance levels on the cold-water coral Desmophyllum pertusum (Lophelia pertusa) from realistic exposure conditions to suspended bentonite, barite and drill cutting particles”PLOS ONE

Dear Dr. Baussant,

Thank you for submitting your manuscript to PLOS ONE. After careful consideration, we feel that it has merit but does not fully meet PLOS ONE’s publication criteria as it currently stands. Therefore, we invite you to submit a revised version of the manuscript that addresses the points raised during the review process.

I apologize for the delay in getting your work reviewed. It has been very difficult to find reviewers due to many things happening, I have now one reviewer's report and I have read through your manuscript and I think, as mentioned by the reviewer, there are some clarifications needed with respect to material and methods and data analysis.please revise according the to comment and resubmitdepending on the changes, I may or may not send it out for review againlooking forward to the revised manuscript

We look forward to receiving your revised manuscript.

Kind regards,

Shashank Keshavmurthy, PhD

Academic Editor

PLOS ONE

Journal Requirements:

3. In your Methods section, please provide additional location information of the collection sites, including geographic coordinates for the data set if available.

"The performance of this study and the writing of the manuscript was financed by Equinor AS. We are particularly grateful for their role in the study design, discussions of the results, decision to publish, discussions and comments on the manuscript made by Tone K. Frost and Ingunn Nilsen from Equinor AS. NORCE Norwegian Research Centre AS contributed also with internal financial support to the final writing of the manuscript."

"The performance of this study and the writing of the manuscript was financed by Equinor AS. We are particularly grateful for their role in the study design, discussions of the results, decision to publish, discussions and comments on the manuscript made by Tone K. Frost and Ingunn Nilsen from Equinor AS. NORCE Norwegian Research Centre AS contributed also with internal financial support to the final writing of the manuscript."

5. Thank you for stating the following in the Financial Disclosure section: 

"This study and the writing of the manuscript was financed by Equinor AS under contract #4590073409. We are particularly grateful for their role in the study design, discussions of the results, decision to publish, discussions and comments on the manuscript made by Tone K. Frost and Ingunn Nilsen from Equinor AS."  

We note that you received funding from a commercial source: Equinor AS

7. PLOS requires an ORCID iD for the corresponding author in Editorial Manager on papers submitted after December 6th, 2016. Please ensure that you have an ORCID iD and that it is validated in Editorial Manager. To do this, go to ‘Update my Information’ (in the upper left-hand corner of the main menu), and click on the Fetch/Validate link next to the ORCID field. This will take you to the ORCID site and allow you to create a new iD or authenticate a pre-existing iD in Editorial Manager. Please see the following video for instructions on linking an ORCID iD to your Editorial Manager account: https://www.youtube.com/watch?v=_xcclfuvtxQ

8. We note that Figure 6 in your submission contain copyrighted images. All PLOS content is published under the Creative Commons Attribution License (CC BY 4.0), which means that the manuscript, images, and Supporting Information files will be freely available online, and any third party is permitted to access, download, copy, distribute, and use these materials in any way, even commercially, with proper attribution. For more information, see our copyright guidelines: http://journals.plos.org/plosone/s/licenses-and-copyright.

a. You may seek permission from the original copyright holder of Figure 6 to publish the content specifically under the CC BY 4.0 license. 

Reviewers' comments:

Reviewer's Responses to Questions

**Comments to the Author**

1. Is the manuscript technically sound, and do the data support the conclusions?

Reviewer #1: Yes

2. Has the statistical analysis been performed appropriately and rigorously? 

Reviewer #1: Yes

3. Have the authors made all data underlying the findings in their manuscript fully available?

Reviewer #1: No

4. Is the manuscript presented in an intelligible fashion and written in standard English?

Reviewer #1: Yes

5. Review Comments to the Author

Reviewer #1: I reviewed the ms entitled “Identification of tolerance levels on the cold-water coral Desmophyllum pertusum (Lophelia pertusa) from realistic exposure conditions to suspended bentonite, barite and drill cutting particles” by Thierry Baussant et al. This study explored the tolerances of a CWC (Desmophyllum pertusum syn. Lophelia pertusa) to suspended particle concentration from the oil industry, particularly drill cutting particles. I am aware that the information about this topic is still scarce, thus, this is an important contribution to the general topic of CWC’s and its vulnerability to offshore oil-drilling operations. Overall, the manuscript is well written, technically sound, and the data support the conclusions. Furthermore, I can see interesting findings, particularly in the tolerance of this species to those additional stressors. I am concerned with some of their approaches in the data analysis and there is some missing information in the way how the replicates were treated (replicates for experimental design, decisions made for pooling data, etc). However, I consider these additions to be minor and easily incorporated into the ms.

The methods are well written and sufficiently detailed as to replicate the experimental work. I would include a table of the ANOVA’s with the F-statistic, degree of freedom, p-value, etc. Also please include the n for all the analysis. It is not clearly stated if the experimental design and data analysis was performed on the actual “nubbins” (pooled data) or at the tank/cone replication. If it was the former, a better description is needed on how did you decide to pool data, What p-value did you use? What considerations were taken into account for pooling or not pooling?Introduction

The introduction is well written and provide with the enough information for the study. It has a clear and stated hypothesis.

Line 91: Should be mg l-1?

Methods

Line 183-189: Please provide more information about the measurement equipment: salinity and temperature with precision and accuracy of the measurement.

Line 147 – line 195: Wrong citation format

Line 206 – 210: Is this a standardize methodology? If so, please provide with a cite for this method.

Coral experiments and measurements

Line 227-230: It is not clear to me how you got dry-weight from B-W, since B-W means to weight the fragments in the water. Also, I understand DW was used for standardization of % growth rate, but also to quantify growth rates? I am a little confused about this. Please clarify.

Line 230: I know surface area is better than DW for standardization of respiration rates. Please provide a reasoning why using DW instead of SA? Can you cite previous work for this?

Data analysis

lines 273 – 277: “replicate tank nested factor was tested using a Kruskal-Wallis analysis (p<0.05), and the data from the replicate tanks were pooled if the assumption of equal distribution was verified”. Please be more specific about this analysis. Did you pool all your fragments from the different cones? or Cones were treated as independent?

Line 286 – 290. The authors state that for growth rate and polyp mortality measurements, “the

ANOVA assumptions were violated, and the effects of DC and BABE treatments were analyzed using a Kruskal–Wallis test on ranks at each sampling time, i.e. T1, T2 (p<0.05)”. Did you apply corrections for multiple comparison? i.e. Bonferroni correction? Also, it is not clear to me what do you mean with “Kruskal–Wallis test on ranks at each sampling time, i.e. T1, T2 (p < 0.05)? T1 and T2 were significantly different? are those values part of the results? The same as for line 290: Were values pooled or not pooled?

Line 264 – check for typo

Line 288 – check for typo

Results and Discussion

Respiration rates: “Under stress conditions, such as with the presence of solid particles in the water, one could expect an increase in respiration rate consequent to the higher metabolic activity resulting from the removal of particles by polyps and mucus secretion”. Could you provide a better explanation on why respiration rates are unaffected under such stressful conditions? Why is this lack of sensitivity?

Polyp mortality:

Line 506 – 510: “In the DC experiment, we found that polyp mortality determination measured during days with addition of food particles and days without food addition was equal (Wilcoxon signed rank test, p=0.103). Hence, data from days of feeding and no feeding were pooled to test differences in polyp mortality resulting from DC treatments at T1 and T2”. As stated above, I am a little bit concern about the procedure of pooling data. Do you have any other previous work that validate your assumption? Normally, a conservative estimate of the p-value for pooling data should be above p>0.25, however, I might be outdated about this procedure. Please provide a reasoning.

Line 531 – 533: I would start the paragraph with the actual value of % polyp mortality found in your study, and then compared to the other studies. I know you referenced the figure 7, but it would be helpful for the reader to find the actual values found in your study written in the paragraph.

Line 551 – 554: It looks speculative. Does not support/explain your results.

Figure 2. What is the n for each treatment? See also comments in the methods section.

6. PLOS authors have the option to publish the peer review history of their article (what does this mean?). If published, this will include your full peer review and any attached files.

Reviewer #1: No

---

## [Author Response · Author response to Decision Letter 0]

28 Dec 2021

Introduction

The introduction is well written and provide with the enough information for the study. It has a clear and stated hypothesis.

Good. Thank you !

Line 91: Should be mg l-1?

Corrected

Methods

Line 183-189: Please provide more information about the measurement equipment: salinity and temperature with precision and accuracy of the measurement.

This information has now been added in the text of the revised version 

CTD model AquaTroll 100, In-Situ Inc, USA

Salinity accuracy: ±0.5-1% of reading, + 1 �S/cm when reading < 80,000 �S/cm and resolution: 0.1 �S/cm;

Temperature accuracy ± 0.1°C and resolution ~ 0.01°C

Line 147 – line 195: Wrong citation format

This has been corrected

Line 206 – 210: Is this a standardize methodology? If so, please provide with a cite for this method.

No, not exactly. As far as I know, there is no standardize methodology for measuring particle concentration in the water but it is common to use weighing on pre-weighted GF/F filters after filtration or turbidity/nephelometry measurements with the use of a standard curve established with a series of preset concentrations. The method is for example used in Larsson et al. (2013) and Baussant et al. (2018). These references are added in the text now.

Coral experiments and measurements

Line 227-230: 1) It is not clear to me how you got dry-weight from B-W, since B-W means to weight the fragments in the water.2) Also, I understand DW was used for standardization of % growth rate, but also to quantify growth rates? I am a little confused about this. Please clarify.

1) Please see paper “Spencer Davies, P. Short-term growth measurements of corals using an accurate buoyant weighing technique. Marine Biology 101, 389–395 (1989). https://doi.org/10.1007/BF00428135” for a full explanation of the calculation of skeleton DW based on skeleton Buoyancy weight. The skeletal weight of the coral is estimated from the buoyant weight in seawater. The mean density of the skeleton has been determined in previous coral work.

2) Yes, this is correct. DW was used for calculation (not standardization)of growth rate from which % growth rate was obtained. Once skeleton DW of all coral nubbins was calculated, the values were used to normalize respiration rate calculated as mgO2 respired/h/gDW skeleton at each time point and in the calculation of % growth rate (%skeleton growth/day during the experimental period from T0 to T1 and T1 to T2) as well.

The text has been slightly modified to reflect these methodological aspects

Line 230: I know surface area is better than DW for standardization of respiration rates. Please provide a reasoning why using DW instead of SA? Can you cite previous work for this?

As far as we know, both ways of standardization of respiration rates are used and accepted. See for example Reynaud and Ferrier-Pages 2019; Dodds et al. 2007; Larsson et al. 2013; Baussant et al. 2017. As the authors of the present paper have normalized respiration rates to DW in previous work, we chose to use the same normalization method in the present article too.

These references have been added to the corresponding line.

Data analysis

lines 273 – 277: “replicate tank nested factor was tested using a Kruskal-Wallis analysis (p<0.05), and the data from the replicate tanks were pooled if the assumption of equal distribution was verified”. Please be more specific about this analysis. Did you pool all your fragments from the different cones? or Cones were treated as independent?

Yes, the fragments of corals from the different cones were pooled after the assumption of equal data distribution in the different cones was verified using the Kruskal-Wallis analysis. The H0 hypothesis tested was that the distribution of all variables measured in corals was the same across the different cones. Below an example of output from the KW analysis test (SPSS statistical software) for respiration data measured in the DC exposure.

The text has been slightly modified to make that clearer.

Line 286 – 290. The authors state that for growth rate and polyp mortality measurements, “the

ANOVA assumptions were violated, and the effects of DC and BABE treatments were analyzed using a Kruskal–Wallis test on ranks at each sampling time, i.e. T1, T2 (p<0.05)”. 1)Did you apply corrections for multiple comparison? i.e. Bonferroni correction? Also, it is not clear to me what do you mean with “Kruskal–Wallis test on ranks at each sampling time, i.e. T1, T2 (p < 0.05)? T1 and T2 were significantly different? are those values part of the results? 2) The same as for line 290: Were values pooled or not pooled?

We have modified the text to clarify better this point. Here is how the text reads now:

1) Prior to analysis, the nested factor “replicate tank” was tested for equal distribution of coral measurement variance across the replicate tanks and, given the few number of coral nubbins in each tank (n=4 to 6), a non-parametric analysis was used. We found no significant difference in coral measurement variance across the different cones (Kruskal-Wallis rank test, p<0.05) and coral data from the different replicate tanks (n=3) of treatment and control were pooled. Then a Kolmogorov-Smirnov was applied to test for Gaussian distribution and the assumption of equal variance (homoscedasticity) for the different treatment groups was verified (Levene’s test) for p>0.05. These assumptions were met for all measurements but for coral growth rate and polyp mortality for which a Kruskal–Wallis rank non-parametric test was applied, tested for each sampling time i.e. for T1 and T2 (p < 0.05) and a Mann-Whitney 2-sample test used as post-hoc analysis. For all other measurements, a repeated ANOVA was applied to test mean differences with treatment concentration as a between-subjects factor and sampling time as a within-subject factor. When the assumption of sphericity was not met (Mauchly’s test, p < 0.05), p-values were adjusted for repeated time measurements using Greenhouse–Geisser epsilon and Hunyh–Feldt epsilon. A post hoc analysis with Tukey’s honest significant difference (Tukey’s HSD) test was applied to identify differences between means in treatment and control groups. The output of the Kruskal-Wallis test is part of the results and now added in the text.

2) The polyp mortality data were also tested for equal distribution between replicate cones and were pooled as commented above in the data analysis section. As polyp mortality was measured during days of feeding and days without feeding, we also pooled polyp mortality from days with and without feeding to increase the number of observations when there was no significant differences or used only days with feeding if there was a significant difference. This is now modified in the text as “Polyp mortality scoring determined during the days with and without feeding were pooled if no significant difference was observed or otherwise only days with feeding were used if there was a significant difference (Wilcoxon signed rank test, p≤0.5).” 

Line 264 – check for typo

Outliers and extreme values were not removed from the overall data analysis, taking into

Corrected in the text

Line 288 – check for typo

the ANOVA assumptions were violated, and the effects of DC and BABE treatments were analyzed using a Kruskal–Wallis rank test for each sampling time separately, i.e. for T1 and T2 (p < 0.05). Corrected in the text

Results and Discussion

Respiration rates: “Under stress conditions, such as with the presence of solid particles in the water, one could expect an increase in respiration rate consequent to the higher metabolic activity resulting from the removal of particles by polyps and mucus secretion”. Could you provide a better explanation on why respiration rates are unaffected under such stressful conditions? Why is this lack of sensitivity?

It is difficult to say but lack of change in respiration rate after particle exposure on deep-sea corals has been reported before as well, even for longer exposure time (see for example Larsson et al., 2013; Baussant et al., 2018). Here the corals were exposed to even shorter exposure (5 days). It has been suggested that the lack of change in respiration rate might be due to optimal coral feeding in laboratory experiments, masking the real effect of stressors on the respiration rates (Reynaud and Ferrier-Pagès, 2019). I do not believe this is a lack senstivity in that measurement since we do measure change when the corals are deprived from food for a long period for example (Larsson et al., 2013; Baussant et al., 2017). 

The text has been slightly adapted in the corresponding section

Polyp mortality:

Line 506 – 510: “In the DC experiment, we found that polyp mortality determination measured during days with addition of food particles and days without food addition was equal (Wilcoxon signed rank test, p=0.103). Hence, data from days of feeding and no feeding were pooled to test differences in polyp mortality resulting from DC treatments at T1 and T2”. As stated above, I am a little bit concern about the procedure of pooling data. Do you have any other previous work that validate your assumption? Normally, a conservative estimate of the p-value for pooling data should be above p>0.25, however, I might be outdated about this procedure. Please provide a reasoning.

I am sorry, we are not aware that a conservative estimate of the p-value for pooling data should be above p>0.25. We have used this pooling approach in two former papers (Baussant et al., 2017 and 2018) with corals in a similar setup and used the same methodology and p value to justify pooling. 

Line 531 – 533: I would start the paragraph with the actual value of % polyp mortality found in your study, and then compared to the other studies. I know you referenced the figure 7, but it would be helpful for the reader to find the actual values found in your study written in the paragraph.

Thank you for this comment. This makes sense. The actual value of % polyp mortality found in your study has been moved at the front of the paragraph and the comparison to other studies at the end of the paragraph of this section.

Line 551 – 554: It looks speculative. Does not support/explain your results.

“However, the coenosarc condition was not checked accurately for these branches at the start of the experiment, so it is difficult to conclude if coenosarc condition at start of the experiment or deterioration during the exposure is explaining the relatively high polyp mortality observed in the DC experiment.”

This sentence was removed and the text of this section was slightly adjusted.

Figure 2. What is the n for each treatment? See also comments in the methods section.

The n was added on the legend of each figure to reflect that the data from each replicate tank for treatment and control group were pooled at each sampling time.

---

## [Editor Report · Decision Letter 1]

12 Jan 2022

Identification of tolerance levels on the cold-water coral Desmophyllum pertusum (Lophelia pertusa) from realistic exposure conditions to suspended bentonite, barite and drill cutting particles

PONE-D-21-17662R1

Dear Dr. Baussant,

We’re pleased to inform you that your manuscript has been judged scientifically suitable for publication and will be formally accepted for publication once it meets all outstanding technical requirements.

Kind regards,

Shashank Keshavmurthy, PhD

Academic Editor

PLOS ONE
---

## [Editor Report · Acceptance letter]

11 Feb 2022

PONE-D-21-17662R1 

Identification of tolerance levels on the cold-water coral *Desmophyllum pertusum (Lophelia pertusa)* from realistic exposure conditions to suspended bentonite, barite and drill cutting particles 

Dear Dr. Baussant:

I'm pleased to inform you that your manuscript has been deemed suitable for publication in PLOS ONE. Congratulations! Your manuscript is now with our production department. 

Kind regards, 

on behalf of

Dr. Shashank Keshavmurthy 

Academic Editor

PLOS ONE